# Transcriptional Signatures of Domestication Revealed through Meta-Analysis of Pig, Chicken, Wild Boar, and Red Junglefowl Gene Expression Data

**DOI:** 10.3390/ani14131998

**Published:** 2024-07-06

**Authors:** Motoki Uno, Hidemasa Bono

**Affiliations:** 1Graduate School of Integrated Sciences for Life, Hiroshima University, 3-10-23 Kagamiyama, Higashi-Hiroshima 739-0046, Japan; m230361@hiroshima-u.ac.jp; 2Genome Editing Innovation Center, Hiroshima University, 3-10-23 Kagamiyama, Higashi-Hiroshima 739-0046, Japan

**Keywords:** *Sus scrofa domesticus*, *Sus scrofa*, *Gallus gallus*, domesticated animals, breeding, gene set enrichment analysis

## Abstract

**Simple Summary:**

Domesticated animals, such as pigs and chickens, have undergone significant genetic changes compared to their wild ancestors. We collected gene expression data from pigs, chickens, wild boars, and red junglefowl and performed a comprehensive analysis. We identified differentially expressed genes by comparing the gene expression patterns of domesticated animals and their wild counterparts. These genes are involved in various biological processes, including immune response, metabolism, and stress response. Notably, domesticated animals exhibited higher expression of genes related to viral resistance and bone weakness, whereas their wild ancestors showed higher expression of genes associated with stress response and energy metabolism. Identifying these differentially expressed genes provides valuable insights into the genetic changes that occurred during the domestication process. Furthermore, these findings have highlighted potential candidate genes that could be targeted in breeding programs to improve the health and productivity of domesticated animals, ultimately contributing to the development of sustainable livestock production and addressing the growing global food demand.

**Abstract:**

Domesticated animals have undergone significant changes in their behavior, morphology, and physiological functions during domestication. To identify the changes in gene expression associated with domestication, we collected the RNA-seq data of pigs, chickens, wild boars, and red junglefowl from public databases and performed a meta-analysis. Gene expression was quantified, and the expression ratio between domesticated animals and their wild ancestors (DW-ratio) was calculated. Genes were classified as “upregulated”, “downregulated”, or “unchanged” based on their DW-ratio, and the DW-score was calculated for each gene. Gene set enrichment analysis revealed that genes upregulated in pigs were related to defense from viral infection, whereas those upregulated in chickens were associated with aminoglycan and carbohydrate derivative catabolic processes. Genes commonly upregulated in pigs and chickens are involved in the immune response, olfactory learning, epigenetic regulation, cell division, and extracellular matrix. In contrast, genes upregulated in wild boar and red junglefowl are related to stress response, cell proliferation, cardiovascular function, neural regulation, and energy metabolism. These findings provide valuable insights into the genetic basis of the domestication process and highlight potential candidate genes for breeding applications.

## 1. Introduction

With the rapid growth of the global population, food shortage has become a serious concern, and it is estimated that the global food demand in 2050 will be approximately 1.5 times that of 2010 [1]. Pigs and chickens are crucial protein sources for humans [2], and their breeding and improvement are essential to address this challenge. Pigs provide meat, whereas chickens supply both meat and eggs, and both are used in pharmaceutical manufacturing [3]. Therefore, expanding our knowledge of pigs and chickens is crucial for achieving a sustainable society.

Pigs were domesticated from wild boars [4], while chickens are presumed to have been domesticated from red junglefowl [5]. Through domestication, these animals have been brought under human control and have been selectively bred for specific traits useful to humans, such as faster growth rates, improved meat quality, and increased litter size [6,7]. This process has significantly altered the animals’ behavior, morphology, and physiological functions. Therefore, we hypothesized that common patterns of gene expression changes associated with domestication might be observed across different species, such as pigs and chickens. However, while the selection for traits beneficial to humans has occurred, excessive selective breeding has led to health issues in domesticated animals and decreased their reproductive capacity [8]. To address these issues, it is crucial to understand the changes in gene expression associated with domestication. Therefore, this study aimed to elucidate the changes in gene expression between domesticated animals and their wild ancestors, seeking to gain insights into the genes and characteristic functions that exhibit altered expression during the domestication process.

Genetic research on domestication and breeding has long focused on mutations in the genome. Quantitative trait locus (QTL) analysis has been used to identify trait-associated loci, such as the NR6A1 locus in pigs, which is associated with the vertebral number [9]. However, QTL analysis has limitations in directly identifying genes related to traits, capturing changes in gene expression, and detecting genes without sequence variation. In contrast, the relationship between changes in gene expression and traits has become apparent, as exemplified by the myostatin gene, where decreased expression leads to increased muscle mass [10]. Transcriptomic analysis is a powerful method for comprehensively identifying domestication-associated changes in gene expression. However, small sample sizes and biases in sampled tissues and breeds make it difficult to identify genes that universally contribute to domestication. Meta-analyses overcome these challenges by integrating data from different breeds, tissues, and growth stages, thereby enabling the elucidation of insights underlying domestication that individual studies cannot reveal [11].

Here, we aimed to identify gene expression changes associated with domestication by collecting gene expression data for pigs, chickens, wild boars, and red junglefowl from public databases and performing a meta-analysis. To the best of our knowledge, this is the first meta-analysis to demonstrate the differences in gene expression between domesticated animals and their wild ancestors, specifically for pigs and wild boars and chickens and red junglefowl. Furthermore, this meta-analysis is the first to integrate multiple studies and identify genes that are commonly differentially expressed across species, both in domesticated animals and their wild ancestors. The meta-analysis revealed that genes related to viral infection defense and leucine uptake were differentially expressed between pigs and wild boars, while genes involved in aminoglycan catabolic processes and the carbon dioxide transport were differentially expressed between chickens and red junglefowl. Furthermore, we identified genes commonly upregulated in domesticated animals, including those involved in the immune response, olfactory learning, and epigenetic regulation. In contrast, genes associated with stress response, cell proliferation, cardiovascular function, and energy metabolism were found to be commonly upregulated in wild ancestors. The insights gained from comparing domesticated animals with their wild ancestors will provide a genetic basis for domestication, which can be combined with technologies such as genome editing to enable more efficient and sustainable livestock breeding. This meta-analysis is expected to capture changes in gene expression that may have been overlooked in individual studies. The findings of this study are expected to contribute to addressing food problems and developing sustainable livestock breeding strategies.

## 2. Materials and Methods

### 2.1. Curation of Gene Expression Data from Public Databases

RNA-seq data were obtained from public databases, the National Center for Biotechnology Information Gene Expression Omnibus (NCBI GEO) [12], and ArrayExpress from European Bioinformatics Institute BioStudies (EBI BioStudies) [13]. For NCBI GEO, a search was conducted using “*Sus scrofa*” and “*Gallus gallus*”. To further refine the search results, the filter “Expression profiling by high throughput sequencing” was applied. In EBI BioStudies’ ArrayExpress, a search was performed with the following parameters: organism set to either “*Sus scrofa*” or “*Gallus gallus*”, technology set to “sequencing assay”, and assay by molecule set to “rna assay” to narrow down the results. Based on search results, manual curation was performed to collect pairs of domesticated animals and their wild ancestors from the same project.

### 2.2. Gene Expression Quantification

FASTQ format files were obtained from the NCBI GEO using the fasterq-dump (version 3.8.0) command from the SRA toolkit [14] based on the respective accession numbers. Quality control of the raw reads was performed using Trim Galore (version 0.6.10) [15]. Transcript quantification was performed using Salmon (version 1.10.2) [16], with Sscrofa11.1 being used as the reference cDNA for pig and wild boar samples and bGalgal1.mat.broiler.GRCg7b being used as the reference cDNA for chicken and red junglefowl samples. The resulting quantified RNA-seq data were expressed as transcripts per million (TPM). Transcript-level TPM values were summarized at the gene level using the tximport (version 1.28.0) [17] package in R.

### 2.3. Calculation of DW-Ratio

Gene expression data were normalized to the DW-ratio. Each letter of DW, respectively, stands for domesticated (D) and wild (W). The DW-ratio was calculated using the following formula:$$DW-ratio=\frac{DTPM+1}{WTPM+1}$$

When calculating the DW-ratio, 1 was added to the TPM values to avoid division by zero and handle genes with zero expression values.

### 2.4. Classification of Differentially Expressed Genes (DEGs)

To evaluate the genes that exhibited changes in expression between domesticated animals and their wild ancestors, all genes were classified into three groups. When the DW-ratio was above a specified threshold, the gene was classified as “upregulated”. In contrast, when the DW-ratio was below a specified threshold, the gene was classified as “downregulated”. Genes that did not fall into either category were classified as “unchanged”. For the upregulated category, thresholds of 1.5-, 2-, 3-, 4-, and 5-fold were tested and a 1.5-fold threshold was adopted. For the downregulated category, thresholds of 1/1.5-, 1/2-, 1/3-, 1/4-, and 1/5- were tested and a 1/1.5-fold threshold was adopted.

### 2.5. Calculation of DW-Scores

To integrate and evaluate the DEGs across different experiments, the DW-score was calculated. The DW-score for each gene was determined by subtracting the number of domesticated–wild pairs in which the gene was classified as “downregulated” from the number of pairs in which it was classified as “upregulated”. The DW-ratio and DW-score were computed using code that had been utilized in a previous study [18].

### 2.6. Gene Set Enrichment Analysis

Based on the obtained DW-score, the top- and bottom-ranked genes were selected, and gene set enrichment analysis was performed using ShinyGO 0.80 [19]. For the pig and wild boar analysis, the species was set to “Pig-Duroc genes Sscrofa11.1”, and the “GO Biological Process” was selected as the Pathway database. All other parameters were maintained at their default settings. For the chicken and red junglefowl comparison, no enriched terms were obtained when using the “Chicken maternal Broiler genes bGalGal1.mat.broiler.GRCg7b” annotation. Therefore, the gene IDs were converted to human gene IDs, and the enrichment analysis was performed using “Human genes GRCh38.p13”.

### 2.7. Commonly Upregulated Genes in Domesticated Animals and Wild Ancestors

To enable cross-species analysis, the gene IDs from each species were converted into human gene IDs (GRCh38). Based on the DW-score, ranked lists of the top- and bottom-ranked genes were generated. The overlap of commonly upregulated genes between domesticated animals and wild ancestors was investigated and visualized using the Venn diagram tool [20].

## 3. Results

### 3.1. Overview of This Study

In this study, we performed a meta-analysis to identify DEGs between domesticated animals and their wild ancestors. The research workflow is illustrated in Figure 1.

First, we manually curated and obtained gene expression data from public databases for pigs, chickens, wild boars, and red jungle fowl. The acquired RNA-seq datasets were processed for gene expression quantification and the calculation of the expression ratio between domesticated animals and their wild ancestors (DW-ratio).

Next, we classified genes as “upregulated”, “downregulated”, or “unchanged” based on their DW-ratio and calculated a DW-score for each gene. Gene set enrichment analysis was conducted using the top-ranked genes based on their DW-score for each combination of domesticated animals and wild ancestors.

Finally, to identify genes that are commonly differentially expressed across different species, we converted gene IDs to human gene IDs and examined the overlap of DEGs between domesticated animals and their wild ancestors.

### 3.2. Curation and Breakdown of RNA-Seq Data Obtained from Public Databases

In this study, we obtained 177 RNA-seq datasets from nine projects in the NCBI GEO [12] and EBI BioStudies [13] databases. To enable the integration and analysis of data from different research projects, microarray data were excluded, and only RNA-seq data were obtained. Additionally, all pairs formed were within single projects to minimize the differences in gene expression between projects. We acquired 46 pairs of pigs and their wild ancestors, wild boars, and 59 pairs of chickens and their presumed wild ancestors, red junglefowl. In the pig–wild boar pairs, muscle tissue was the most common tissue examined, being found in 18 pairs, followed by the spleen in 12 pairs and the brain in 11 pairs. In pairs of chickens and red junglefowl, the brain was the most common, with 19 pairs, followed by the testes with 10 pairs and the eyes with 8 pairs (Figure 2). The metadata of the curated dataset, including the accession IDs, breeds, sampled tissues, ages, and sexes of the collected RNA-seq data, are presented in Appendix A.

### 3.3. Characterization of DEGs and Enrichment Analysis of Pigs and Boars

After gene expression quantification using Salmon [16] (Appendix A), the expression ratio between domesticated animals and their wild ancestors (DW-ratio) was calculated. A 1.5-fold threshold was chosen for both upregulated and downregulated genes to capture a comprehensive set of DEGs while minimizing the omission of potentially relevant candidates. For each sample, genes with a DW-ratio ≥ 1.5 were classified as “upregulated”, those with a DW-ratio less than or equal to 1/1.5 were classified as “downregulated”, and the remaining genes were classified as “unchanged”. For each gene, a DW-score was calculated by subtracting the number of samples in which the gene was classified as “downregulated” from the number of samples in which it was classified as “upregulated”. By ranking the genes based on their DW-score, a list of DEGs between domesticated animals and their wild ancestors was obtained. The rankings of DEGs at different thresholds are shown in Appendix A. To investigate whether there were any functional biases in the list of DEGs, enrichment analysis was performed using ShinyGO 0.80 [19].

For the pig–wild boar pairs, enrichment analysis was conducted using DEGs (218 and 239 genes upregulated in pigs and wild boars, respectively (Figure 3A, Appendix A). The threshold for the number of genes used in enrichment analysis was determined based on the DW-scores that showed biases in the Gene Ontology (GO) term function, corresponding to approximately 1% of the total number of genes in the reference genome. The enrichment analysis of the gene list upregulated in pigs revealed a significant enrichment of GO terms related to viral infection defense, such as “Negative regulation of viral genome replication” and “Positive regulation of interferon-beta production” (Figure 3B, Appendix A). The enrichment analysis results for genes upregulated in wild boars showed significant enrichment of GO terms related to leucine uptake, such as “Leucine import across plasma membrane” and “L-leucine import across plasma membrane” (Figure 3C, Appendix A).

### 3.4. Characterization of DEGs and Enrichment Analysis in Chickens and Red Junglefowl

An enrichment analysis was performed for chickens and their wild ancestor, red junglefowl, based on the obtained DW-score. In this case, no functional enrichment was observed; therefore, the gene IDs were converted to human gene IDs, and an enrichment analysis was performed using 206 and 200 chicken and red junglefowl genes, respectively (Figure 4A, Appendix A). The enrichment analysis results for genes upregulated in chickens revealed two terms: “Aminoglycan catabolic process” and “Carbohydrate derivative catabolic process” (Figure 4B, Appendix A). The GO terms “Carbon dioxide transport”, “Oxygen transport”, and “Gas transport” were the top-ranked terms for genes upregulated in red junglefowl (Figure 4C, Appendix A).

### 3.5. Common DEGs in Domesticated Animals and Their Wild Ancestors

To investigate whether genes were commonly differentially expressed across different domesticated species, the gene IDs of each species were converted into human gene IDs (Appendix A). The overlap of DEGs was examined between domesticated animals (pigs and chickens) and their wild ancestors (wild boars and red junglefowl) (Figure 5). When examining the overlap between the top 240 upregulated genes in pigs and the top 206 upregulated genes in chickens (Appendix A), we found that 10 genes were upregulated in both pigs and chickens. These genes were *EZH2*, *IFI6*, *PRTFDC1*, *CEP20*, *CMPK2*, *CX3CR1*, *USP18*, *NID1*, *CXCL13*, and *PHF11* (Table 1). When examining the overlap between the top 206 and 200 genes upregulated in wild boars and red junglefowl, respectively (Appendix A), we found that seven genes were upregulated in both wild boars and red junglefowl: *TAGLN*, *POPDC2*, *MAFF*, *HSPB8*, *SCN1B*, *NT5C1A*, and *BAG3* (Table 2).

## 4. Discussion

In this study, the gene expression data of domesticated animals and their wild ancestors were manually curated and obtained from public databases [12,13]. Ultimately, 46 pig–wild boar pairs and 59 chicken–red junglefowl pairs were obtained. Using these data, a meta-analysis was performed to identify DEGs between domesticated animals and their wild ancestors. To characterize DEGs, enrichment analysis was conducted for each combination of domesticated animals and their wild ancestors. Furthermore, to identify genes that were commonly differentially expressed across different species, all gene IDs were converted to human gene IDs. Subsequently, the DEGs shared between domesticated animals, pigs and chickens, and between their wild ancestors, wild boars and red junglefowl, respectively, were investigated. The list of DEGs included several genes that have been reported to be involved in the traits of domesticated animals. This suggests that the gene expression dataset and meta-analysis method used in this study accurately reflect the scientific knowledge regarding gene expression differences between domesticated animals and their wild ancestors.

The list of genes upregulated in pigs included several genes previously reported to be associated with pig traits, such as muscle development and fat accumulation, including *IGF2BP3*, ENSSSCG00000035293 (*IGF2*), *NCAPG*, *FGF11*, and *FABP4*. *IGF2* is a well-known QTL associated with muscle growth and back-fat thickness in pigs [21,22], and its expression level is correlated with skeletal muscle mass [23]. Additionally, *IGF2BP3* is involved in regulating *IGF2* expression [24], suggesting that it may play a role in controlling skeletal muscle growth in pigs [25]. *NCAPG*, which is essential for maintaining chromosome structure [26], is associated with traits such as body size and muscle development in various domesticated animals [27,28,29,30,31,32,33,34]. *FDF11* has been suggested to regulate adipocyte differentiation through the expression of PPARγ, a key transcription factor controlling adipocyte differentiation [35]. *FABP4*, which binds to free fatty acids with high affinity and functions as a lipid transport protein [36], is upregulated in muscle-specific PPARγ-overexpressing pigs and is thought to be involved in intramuscular fat deposition [37]. Moreover, among the genes upregulated in pigs, genes involved in viral or bacterial infections, such as *IFI44L*, *IL33*, and *RSAD2*, were ranked at the top of the list. Furthermore, when enrichment analysis was performed using 218 genes upregulated in pigs, GO terms related to viral infection defense, such as “Negative regulation of viral genome replication” and “Positive regulation of interferon-beta production,” were ranked among the top terms. Notably, the expressions of sensor proteins that bind to viral nucleic acids, such as *RIGI*, *IFIH1* (encoding MDA5), *DHX58* (encoding LGP2), *RIF2AK2* (encoding PKR), *OAS1*, and *OAS2* [38,39,40], were found to be elevated. These findings suggest that during their domestication process, pigs with higher resistance to viruses and pathogenic bacteria were selected. In particular, the upregulated genes encoding several sensor proteins that bind to viral nucleic acids suggest the possibility that individuals exhibiting early-stage resistance to infections were selected to minimize the spread of infection.

The enrichment analysis using 239 genes upregulated in wild boars revealed that GO terms related to leucine uptake, such as “Leucine import across plasma membrane” and “L-leucine import across plasma membrane”, were the most significantly enriched. Like other animals, leucine is also an essential amino acid for pigs [41] and is known to play a crucial role in muscle synthesis. Moreover, leucine has been suggested to promote muscle growth by enhancing mTORC1 activation and its downstream effectors [42]. Furthermore, leucine supplementation suppresses obesity in obese mice fed a high-fat diet [43]. In addition, leucine supplementation may improve leptin sensitivity and suppress appetite [44]. Genes upregulated in wild boars can also be considered downregulated in pigs. Therefore, active leucine uptake in wild boars, which is impaired in pigs, may contribute to fat accumulation and appetite in pigs.

When enrichment analysis was performed on chicken and red junglefowl pairs using chicken GO annotations, no enriched functions were observed. Therefore, when chicken gene IDs were converted to human gene IDs and enrichment analysis was performed using human GO annotations, enriched functions were obtained for both chicken and red junglefowl. This may indicate that current genetic research on chickens is insufficient. Enrichment analysis using a list of 206 upregulated genes that were upregulated in chickens yielded two terms: “Aminoglycan catabolic proc.” and “Carbohydrate derivative catabolic process”. Selection for rapid growth rates resulted in the skeletal development of broilers not keeping pace with their body weight gain, leading to fractures and locomotor disorders [8]. Aminoglycans, a more general term for glycosaminoglycans, promote bone and joint healing [45]. Research is also being conducted to promote the structural development of bones and joints by incorporating glycosaminoglycans into feed [46]. The enrichment of the “Aminoglycan catabolic proc.” term in the enrichment analysis using the gene set upregulated in chickens may reflect the weakness of chicken bones and locomotor disorders.

Using the 200 genes that were upregulated in red junglefowl, enrichment analysis revealed that the GO terms “Carbon dioxide transport”, “Oxygen transport”, and “Gas transport” were highly enriched. These GO terms include genes such as *HBA2*, *HBG2*, *HBA1*, *HBE1*, *HBG1*, *HBD*, and *HBB* (Appendix A). It is known that chickens inhabiting high-altitude regions have higher hemoglobin levels than those living at lower altitudes [47]. These terms and genes may indicate that the red junglefowl possesses a tolerance to hypoxia. Additionally, these results may reflect the activity levels of the red junglefowl compared to domesticated chickens. Fast-growing chickens have been shown to exhibit lower frequencies of various behaviors compared to slow-growing chickens [48]. Such behavioral differences are believed to be a result of selective breeding that suppresses energy expenditure on activities other than growth. Therefore, the results observed in red junglefowl, which move more frequently in natural environments, can be considered in this context. However, it should be noted that the data used for the analysis did not include information on the altitude or environment of the individuals from which the samples were taken.

We examined the overlap of the DEGs among the domesticated animals; a total of 10 genes (*IFI6*, *CX3CR1*, *USP18*, *CXCL13*, *PHF11*, *CMPK2*, *PRTFDC1*, *CEP20*, *EZH2*, and *NID1*) were upregulated in pigs (240 genes) and chickens (206 genes). In contrast, seven genes (*MAFF*, *HSPB8*, *BAG3*, *TAGLN*, *POPDC2*, *SCN1B*, and *NT5C1A*) were commonly upregulated in wild boars (206 genes) and red junglefowl (200 genes). The upregulated genes in the domesticated animals were involved in various biological processes, including immune response (*IFI6* [49], *CX3CR1* [50], *USP18* [51], *CXCL13* [52], *PHF11* [53], *CMPK2* [54]), olfactory learning (*PRTFDC1* [55]), epigenetic regulation (*EZH2* [56]), cell division (*CEP20* [57]), and extracellular matrix (*NID1* [58]). These changes in gene expression may contribute to the adaptation of the immune system, changes in olfaction, the regulation of growth and development, and morphological changes associated with domestication. Domesticated animals are reared in confined spaces and high densities to make them easier to manage. As a result, when some individuals become infected, the infection spreads more easily compared to wild ancestors. Additionally, it is thought that the selection of individuals with strong resistance to infections has influenced the observed common upregulation of multiple genes related to immune response in domesticated animals. In contrast, the upregulated genes in the wild counterparts were related to stress response (*MAFF* [59], *HSPB8* [60], *BAG3* [60]), cell proliferation and differentiation (*TAGLN* [61]), cardiovascular function (*POPDC2* [62]), neural regulation (*SCN1B* [63]), and energy metabolism (*NT5C1A* [64]). Animals living in wild environments are constantly exposed to numerous stressors. In contrast, domesticated animals are raised in environments where various stressors are eliminated to enhance economic traits. The common upregulation of multiple stress response genes in wild ancestors likely reflects these environmental differences and suggests that wild ancestors have maintained physiological functions necessary for adaptation to natural environments. Furthermore, these DEGs were upregulated in domesticated animals and their wild ancestors, suggesting a strong association with the domestication process. This indicates that these genes may serve as universal candidates for breeding applications.

This study has several limitations that should be taken into consideration. First, the selection of DEGs was not based on statistical methods, necessitating caution in interpreting the results. Future studies should employ rigorous statistical approaches to identify DEGs with higher confidence. Second, potential biases may have been introduced due to undisclosed metadata. The lack of complete and accurate metadata for some datasets could have influenced the interpretation of the results and the comparison of gene expression patterns across different studies. Additionally, the metadata on chicken breeds are limited, making it difficult to interpret the results. Efforts should be made to ensure the transparency and completeness of metadata in public databases to facilitate more reliable meta-analyses. Third, the reference transcriptomes used for the wild ancestors (wild boar and red junglefowl) were based on their domesticated counterparts (pig and chicken) due to the lack of well-annotated data. This limitation may have affected the accuracy of gene expression quantification and the identification of species-specific genes. Future research will require high-quality reference data for the wild ancestors to enable accurate comparisons between domesticated animals and their wild ancestors. Despite these limitations, this meta-analysis provides a valuable starting point for understanding the gene expression changes associated with domestication and highlights the need for further research to address these challenges and validate our findings.

To the best of our knowledge, this study is the first meta-analysis to demonstrate differences in gene expression between these domesticated animals and their wild ancestors. This is the first meta-analysis to identify genes that are commonly differentially expressed across different species in both domesticated animals and their wild ancestors. The insights gained from this meta-analysis provide a valuable foundation for understanding the genetic basis of the domestication process and highlighting potential candidate genes for breeding applications.

## 5. Conclusions

In conclusion, this meta-analysis identified DEGs associated with the domestication process. These genes are involved in traits such as muscle development, fat accumulation, viral resistance, and bone weakness in domesticated animals. Genes commonly upregulated in domesticated animals were related to the immune response, olfactory learning, epigenetic regulation, cell division, and the extracellular matrix, whereas those upregulated in wild ancestors were associated with stress response, cell proliferation, cardiovascular function, neural regulation, and energy metabolism.

The findings of this study provide valuable insights into the genetic basis of domestication and highlight potential candidate genes for breeding applications. The DEGs identified in this study may contribute to the adaptation of domesticated animals to their environment and the development of desired traits. Furthermore, the genes commonly upregulated across different domesticated species suggest a strong association with the domestication process and their potential as universal candidate genes for breeding. In future research, candidate genes for breeding should be identified using a wider range of species, and the possibility of applying these candidate genes to efficient breeding programs should be explored using techniques such as genome editing.

## Figures and Tables

**Figure 1 animals-14-01998-f001:**
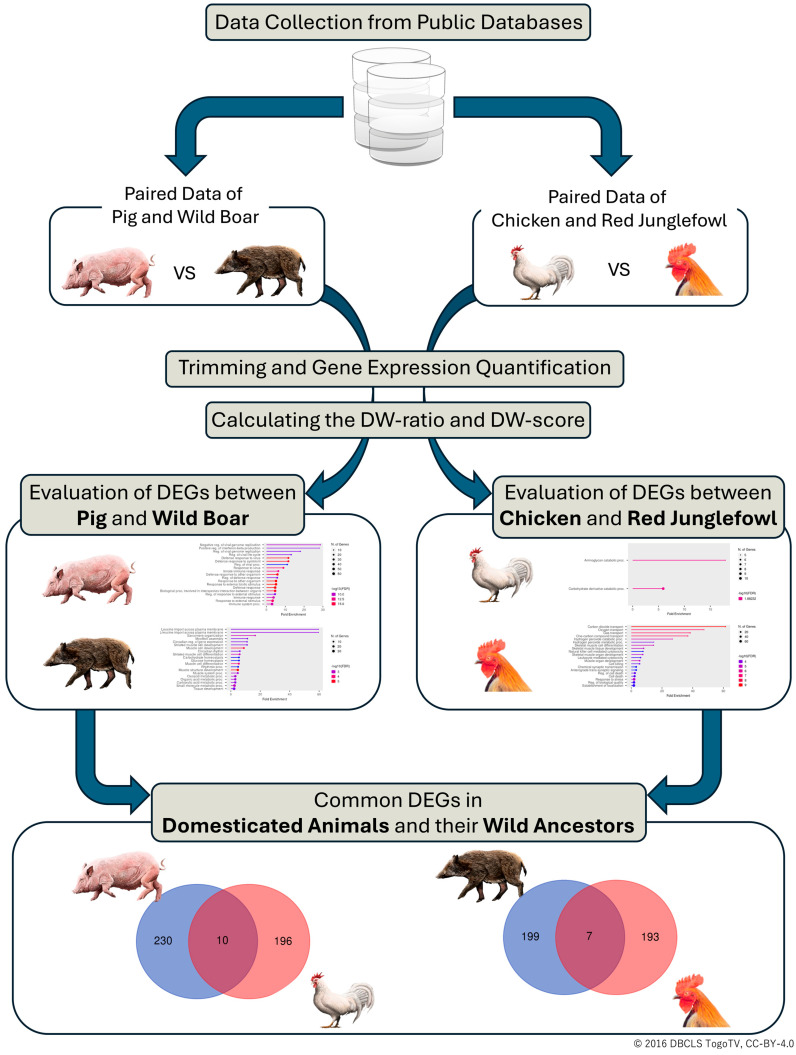
The workflow of this study. The gene expression data of domesticated animals and their wild ancestors were obtained from public databases. The datasets were processed to quantify gene expression and calculate the expression ratio between domesticated animals and wild ancestors (DW-ratio). Genes were classified based on their DW-ratio, and a DW-score was calculated for each gene. Gene set enrichment analysis was performed using top-ranked genes for each domesticated–wild pair. Gene IDs were converted to human gene IDs to identify commonly differentially expressed genes across species. DEG, differentially expressed gene; DW, domestic–wild.

**Figure 2 animals-14-01998-f002:**
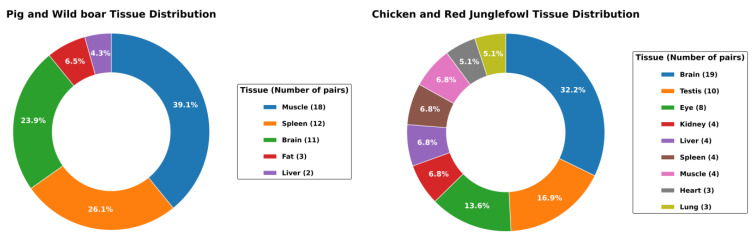
Tissue distribution of the curated RNA-seq data pairs. A total of 46 pig–wild boar pairs and 59 chicken–red junglefowl pairs of RNA-seq data were obtained. The left pie chart shows the proportion of pairs for each tissue type in the pig–wild boar pairs, with muscle tissue having the highest proportion. The right pie chart represents the tissue distribution in the chicken–red junglefowl pairs, with brain tissue having the highest proportion.

**Figure 3 animals-14-01998-f003:**
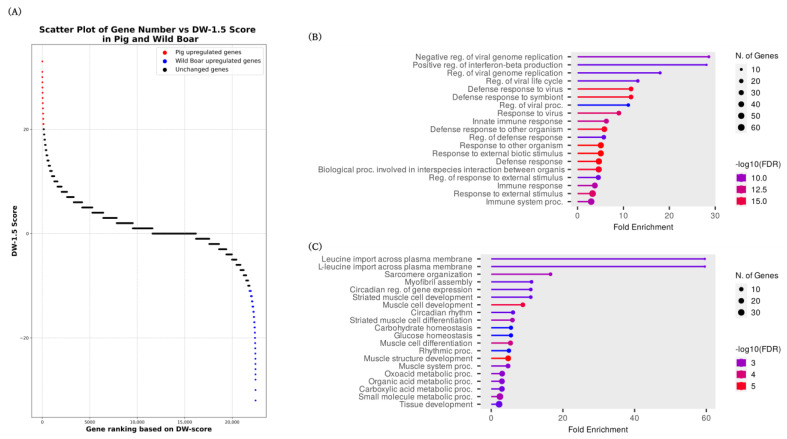
A scatter plot of the DW-score and gene set enrichment analysis of genes upregulated in pigs and wild boars. (**A**) The scatter plot shows the DW-score of each gene, with red dots representing genes upregulated in pigs (positive DW-score values) and blue dots representing genes upregulated in wild boars (negative DW-score values). (**B**) Gene set enrichment analysis for genes upregulated in pigs. The analysis was performed using the top 218 genes based on their DW-scores. (**C**) Gene set enrichment analysis for genes upregulated in wild boars. The analysis was performed the top 239 genes based on their DW-scores.

**Figure 4 animals-14-01998-f004:**
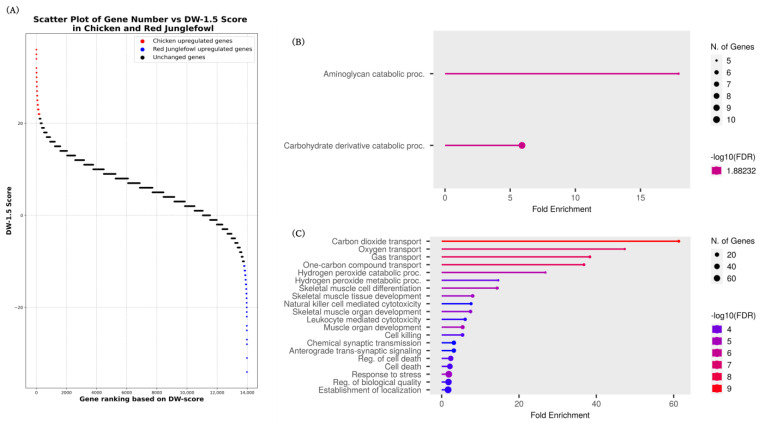
Scatter plot of the DW-score and gene set enrichment analysis of upregulated genes in chickens and red junglefowl. (**A**) The scatter plot shows the DW-score of each gene, with red dots representing genes upregulated in chickens (positive DW-score values) and blue dots representing genes upregulated in red junglefowl (negative DW-score values). (**B**) Gene set enrichment analysis for genes upregulated in chickens. The analysis was performed using the top 206 genes based on their DW-scores. (**C**) Gene set enrichment analysis for genes upregulated in red junglefowl. The analysis was performed with the top 200 genes based on their DW-scores.

**Figure 5 animals-14-01998-f005:**
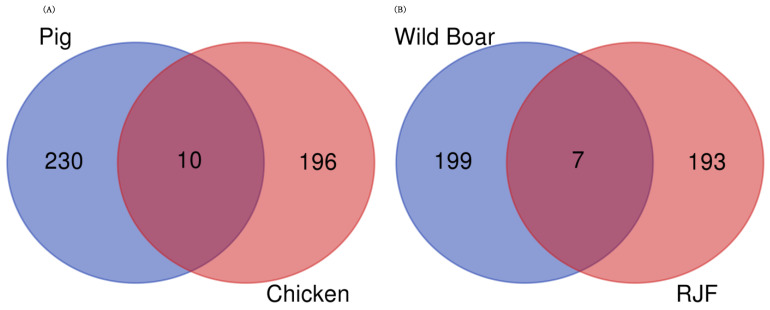
Upregulated genes in common among domesticated animals (pig and chicken) and their wild ancestors (wild boar and red junglefowl). (**A**) A Venn diagram showing the overlap between the top upregulated genes in pigs (240 genes) and chickens (206 genes). Ten genes were found to be upregulated in both domesticated animals. (**B**) A Venn diagram showing the overlap between the top upregulated genes in wild boars (206 genes) and red junglefowl (RJF) (200 genes). Seven genes were found to be upregulated in both wild ancestors.

**Table 1 animals-14-01998-t001:** Upregulated genes in common between domesticated animals (pigs and chickens).

Gene ID	Gene Name	Description
ENSG00000106462	*EZH2*	Enhancer of zeste 2 polycomb repressive complex 2 subunit
ENSG00000126709	*IFI6*	Interferon alpha inducible protein 6
ENSG00000184979	*USP18*	Ubiquitin specific peptidase 18
ENSG00000099256	*PRTFDC1*	Phosphoribosyl transferase domain containing 1
ENSG00000133393	*CEP20*	Centrosomal protein 20
ENSG00000116962	*NID1*	Nidogen 1
ENSG00000156234	*CXCL13*	C-X-C motif chemokine ligand 13
ENSG00000134326	*CMPK2*	Cytidine/uridine monophosphate kinase 2
ENSG00000168329	*CX3CR1*	C-X3-C motif chemokine receptor 1
ENSG00000136147	*PHF11*	PHD finger protein 11

**Table 2 animals-14-01998-t002:** Upregulated genes in common between wild ancestors (wild boars and red junglefowl).

Gene ID	Gene Name	Description
ENSG00000149591	*TAGLN*	Transgelin
ENSG00000121577	*POPDC2*	Popeye domain containing 2
ENSG00000185022	*MAFF*	MAF bZIP transcription factor F
ENSG00000152137	*HSPB8*	Heat shock protein family B (small) member 8
ENSG00000105711	*SCN1B*	Sodium voltage-gated channel beta subunit 1
ENSG00000116981	*NT5C1A*	5′-nucleotidase, cytosolic IA
ENSG00000151929	*BAG3*	BAG cochaperone 3

## Data Availability

The data presented in this study are publicly available at FigShare (https://doi.org/10.6084/m9.figshare.25793967 accessed on 20 May 2024).

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
