# Peer review of "Transcriptional Signatures of Domestication Revealed through Meta-Analysis of Pig, Chicken, Wild Boar, and Red Junglefowl Gene Expression Data"

_animals, 2024, doi:10.3390/ani14131998_

Round 1

Reviewer 1 Report

Comments and Suggestions for Authors

Using public RNA sequencing data from two domestic species, the manuscript describes the meta-analysis of chickens and pigs to identity the differentially expressed genes by comparing gene expression patterns of domesticated animals and their wild counterparts.

The following comments need to address for further evaluation.

1.       According to section 3.2, authors mentioned 9 projects to obtain 177 RNA-seq datasets. Please provide the accession or data identifiers from nine projects and present in Supplementary Table 1 &2 including published citation.

2.        According to Line 190: sex should be included in Supplementary Table 1 &2.

3.       Table 2 same SSR run ID (SRR10143535) was found in column A (domestic) and B (wild type Red junglefowl) and this will affect the accuracy of Meta- analysis. Redo the analysis by choosing another SSR run ID from domestic chicken public database.

4.       There were 17 SSR ID (SRR10143525,SRR10143537,SRR10143541,SRR10143542,SRR10143543,SRR10143544,SRR10143545,SRR10143549,SRR10143552,SRR20219403,SRR20219404,SRR20219405,SRR23370830,SRR23370833,SRR23370836,SRR23370839,SRR23370842) chosen for TM analysis in chicken (see Supplementary Table 4)  and why these ID were missing in Supplementary Table 2 for comparison? If these data included for analysis, detail information should be included in Supplementary Table2.

5.       Gene names should be included in the result of TM analysis Supplementary Table 3 & 4 for gene verification in differentially expressed genes.

6.       Human gene should replace by chicken gene ID in Supplementary Table 6, 11 and 12.

Reviewer 2 Report

Comments and Suggestions for Authors

Dear authors, I would like to state that I am pleased to evaluate your article. I would like to express my criticisms on the following points in your article:

Introduction: 
In the introduction, I would expect the hypothesis and purpose of the study to be adequately discussed with relevant literature and to emphasize why the study should be done. However, the paragraphs in the introduction section are written meaninglessly disconnected from each other. In this section, it should have been emphasized why the study was important for the species studied. In addition, the study examined different species and different genes of each species, and yet the results were not discussed sufficiently, reducing the quality of the study.

Materials and Method:

Not enough information was given about the breeds used in the study. For example, what are the characteristics of the chicken breeds used? Do breeds other than the Red Jungle Fowl have an important role in the domestication process of chickens? Or has an intense selection been made on these breeds? I would have expected this scenario to have been explained.   Results and Discussion: According to the projects’ RNAseq results in the NCBI database, the authors tried to find out the differences of some genes between the origin species and the domesticated breeds. The genes reported in the conclusion of the study are genes that have been studied in previous studies and whose regulation status has been reported, as cited by the authors. Therefore, the discussion on the relevant genes only refers to the regulation status of the genes in the original species and after domestication.
The discussion part also refers the previous study results about the regulation of the studied gene which has been also written at the cited study. However, I was expecting to see some original discussion paragraphs about the reasons of the evolution of these genes through the years across species. For my opinion the discussion part needs to be revised.

Comments on the Quality of English Language

The manuscript was written in an understandable English. However, a proof check service is still recommended.

Reviewer 3 Report

Comments and Suggestions for Authors

The manuscript is written well, few edits for english language may be done. Manuscript may be accepted for publication.

Comments on the Quality of English Language

Can be improved.

Author Response

Thank you very much for your comments. English was checked again for better manuscript.

Round 2

Reviewer 1 Report

Comments and Suggestions for Authors

Authors had answered the questions and made the correction accordingly and appropriately. 

Reviewer 2 Report

Comments and Suggestions for Authors

Dear authors, thank you for your effort for the revisions.

Comments on the Quality of English Language

Still needs English proof check.